# Optimizing Intrusion Detection Systems in Three Phases on the CSE-CIC-IDS-2018 Dataset

Surasit Songma [1,*], Theera Sathuphan [2] and Thanakorn Pamutha [3]

1  Department of Information Technology, Faculty of Science and Technology, Suan Dusit University, Bangkok 10300, Thailand
2  Faculty of Computer Science, Ubon Ratchathani Rajabhat University, Ubonratchathani 34000, Thailand; theera.s@ubru.ac.th
3  Faculty of Science Technology and Agriculture, Yala Rajabhat University, Yala 95000, Thailand; thanakorn.p@yru.ac.th
*  Correspondence: surasit_son@dusit.ac.th; Tel.: +66-81-3723218

**Abstract:** This article examines intrusion detection systems in depth using the CSE-CIC-IDS-2018 dataset. The investigation is divided into three stages: to begin, data cleaning, exploratory data analysis, and data normalization procedures (min-max and Z-score) are used to prepare data for use with various classifiers; second, in order to improve processing speed and reduce model complexity, a combination of principal component analysis (PCA) and random forest (RF) is used to reduce non-significant features by comparing them to the full dataset; finally, machine learning methods (XGBoost, CART, DT, KNN, MLP, RF, LR, and Bayes) are applied to specific features and preprocessing procedures, with the XGBoost, DT, and RF models outperforming the others in terms of both ROC values and CPU runtime. The evaluation concludes with the discovery of an optimal set, which includes PCA and RF feature selection.

**Keywords:** intrusion detection system; machine learning techniques; exploratory data analysis; performance evaluation; feature selection; CSE-CIC-IDS-2018 dataset; three-phase models

## 1. Introduction

In today's world, the Internet has become an invaluable tool, effortlessly integrated into human life. People all over the world utilize it as a communication and information exchange medium. Information and communication technology (ICT) is essential in both business and daily life. However, in the age of big data, cyber-attacks on ICT systems have become increasingly sophisticated and broad, making network risks a key issue in modern life. Malicious attacks are continually developing, emphasizing the critical need for improved network security solutions. Given the world's growing reliance on digital technologies such as computers and the Internet, building safe and reliable programs, frameworks, and networks that can withstand these attacks is a critical task [1,2].

Intrusion detection systems (IDS) are critical for protecting computer networks. They effectively recognize and respond to security threats. Intrusion is used to detect irregularities in network traffic to improve security. Detection accuracy, detection times, false alarm alerts, and the identification of unknown assaults are currently issues for IDS technology [3]. They are classified into three types: signature-based systems, anomaly-based systems, and hybrid systems. Anomaly-based systems can detect unknown hostile actions by recognizing deviations from a model based on typical behavior, whereas signature-based systems can identify known assaults by employing established signatures. Signature-based systems, on the other hand, have a high rate of false alarms [4]. Existing anomaly intrusion detection systems have accuracy problems. Certain datasets lack network traffic diversity and volume, others lack diverse or recent attack patterns, and still others lack crucial feature set metadata. The hybrid IDS, which includes both anomaly-based and misuse-based

IDSs, proved to be a more robust and effective solution. Network intrusion detection systems (NIDS) are critical in resolving security issues. NIDS monitors network traffic for unusual activity, and then analyzes the data to discover security breaches such as invasions, misuse, and anomalies. NIDS must deal with difficulties like large data dimensionality and high traffic volumes [5]. While many research projects have used machine learning techniques, approaches which are useful in NIDS, they have limits when confronted with large amounts of network data. Feature selection (FS) has become widely used in selecting relevant features for building strong models. It has significantly influenced the efficiency and performance of IDS models [6]. As a result, three critical aspects of NIDS development are preprocessing, feature reduction, and classifier methods. Nonetheless, network intrusion detection systems encounter issues such as managing massive amounts of data, high false alarm rates, and skewed data.

Machine learning techniques (ML) have been used widely. They have been used in the field of information security in recent years. ML have found widespread application in network security during the last two decades [7]. ML approaches are becoming more popular as a method of spotting anomalies [8]. ML includes automating the process of learning from examples. It is used to build models that distinguish between regular and aberrant classes [9].

The CSE-CIC-IDS-2018 dataset is used in this study to investigate the complexities of IDS, with the goal of addressing the critical concerns connected with the complexity and resource demands of IDS in the field of cybersecurity. The study is divided into three stages: initial data preparation via data cleaning, exploratory data analysis, and data normalization; subsequent reduction of non-significant features via a combination of principal component analysis (PCA) and random forest (RF) to improve processing speed and reduce model complexity; and application of various machine learning algorithms, with XGBoost, decision trees (DT), and random forest emerging as top performers based on ROC. The work closes with the discovery of an ideal feature set by PCA and RF feature selection, providing a viable way to improve the efficiency and accuracy of intrusion detection systems and thus bringing valuable insights to the cybersecurity arena.

The goal of this study was to find the most effective classifier by methods for preprocessing and feature selection translated into machine learning approaches that are extensively used by us in intrusion detection systems. Popular classification algorithms such as extreme gradient boosting (XGBoost), classification and regression trees (CART), decision tree (DT), k-nearest neighbors (KNN), multilayer perceptron (MLP), random forest (RF), logistic regression (LR), and naïve Bayes (Bayes) are included. The evaluation of performance encompasses several dimensions as nine important criteria: In k-fold cross-validation, accuracy, precision, recall, F1 score, PCC/BA, MCC, ROC, and average were calculated. Classification, central processing unit (CPU) time, and model size were also explored.

The following are the main contributions of this study:

- Investigation of large amounts of data linked with harmful network activity;
- Identification of feature dimensions influencing classification performance in a labeled dataset with both benign and malicious traffic, resulting in improved detection accuracy;
- Use of the CSE-CIC-IDS-2018 dataset for NIDS and testing of seven different machine learning classifiers and scripts for identifying various sorts of assaults;
- In general, researchers frequently work with incomplete data. In contrast, this study uses all accessible DDoS data in the experiment, correlating with reality by adopting the concept of data imbalance;
- Presenting various performance assessments has many elements. Furthermore, the evaluation considers CPU processing time, which is an important component in intrusion detection, as well as the size of the experimentally obtained model, which has the possibility for future extension.

The rest of the paper is structured as follows: Section 2 describes the research sequence, as well as the research concept and process; the methodology and proposed framework

are described in Section 3; the experimental setup is described and defined in Section 4; the experiments and related discussions are presented in Section 5; finally, Section 6 concludes the essay by discussing the model's strengths and flaws and suggesting future study directions.

## 2. Related Work

There are very few datasets for network intrusion detection compared with datasets for malicious code. KDD CUP 99 (KDD) is the most widely used dataset for the evaluation of IDS. Numerous studies on ML-based IDS have been using KDD or the upgraded versions of KDD. In this work, we develop an IDS model using CSE-CIC-IDS-2018, a dataset containing the most up-to-date common network attacks [10]. The Canadian Institute for Cybersecurity's CSE-CIC-IDS-2018 dataset incorporates the concept of profiles. The most recent edition of this dataset provides versatility, allowing both agents and individuals to generate network events. These profiles can be applied to a variety of network protocols and topologies. Furthermore, the dataset has been updated by adding the standards used in the development of CIC-IDS-2017. In addition to meeting the necessary requirements, it provides the following benefits: minimum duplicate data, nearly no unclear information, and the dataset is already in CSV format, making it ready for use without further processing [11].

As data dimensionality grows, feature selection has become a critical preprocessing step in the development of intrusion detection systems. Feature selection entails removing irrelevant and superfluous features and picking the optimal subset that best characterizes patterns in various classes. There are various advantages to feature selection. It reduces feature dimensionality, which leads to better algorithm performance. By removing redundant, irrelevant, or noisy data, it improves data efficiency and thus learning technique performance. It also improves the correctness of the output model and aids in understanding the underlying operations that generated the data [12,13].

Following a study of relevant documents and research articles, it was discovered that several studies used machine learning techniques in conjunction with the CEC-CIC-IDS-2018 dataset to detect intrusions. The following is an overview of these findings, shown in Table 1.

**Table 1.** An overview of research on network intrusion detection methods and findings.

| Study | Methodology/Findings |
| --- | --- |
| S. Ullah. et al. [14] | Compares machine learning algorithms (RF, Bayes, LR, KNN, DT) and feature selection by RF (30 features) using CSE-CIC-IDS-2018 dataset. DT yielded the best results. |
| M. A. Khan. [15] | Develops HCRNNIDS, a hybrid convolutional recurrent neural network-based NIDS, and compares it with machine learning algorithms (DT, LR, XGBoost) and feature selection by RF (30 features) using CSE-CIC-IDS-2018 dataset. HCRNNIDS showed superior results. |
| J. Kim. et al. [10] | Discusses IDS models using various machine learning algorithms (ANN, SVM, CNN, RNN) and finds that CNN outperforms traditional techniques when applied to CSE-CIC-IDS-2018 dataset. |
| R. Qusyairi. et al. [3] | Proposes an ensemble learning technique incorporating LR, DT, and gradient boosting after comparisons with single classifiers, using the CSE-CIC-IDS-2018 dataset. Identified 23 significant traits out of 80. |

**Table 1.** *Cont.*

| Study | Methodology/Findings |
|---|---|
| S. Chimphlee. et al. [4] | Focuses on IDS using the CSE-CIC-IDS-2018 dataset, employs data preprocessing, feature selection, and seven classifier machine learning algorithms (including MLP and XGBoost). MLP provided the most successful outcomes. |
| A. Padmashree. et al. [16] | Addresses the industrial revolution's IoT security challenges by offering a robust model with efficient feature selection, preprocessing, and DT-PCRFE for increased security. The model achieves a stunning 99.2% accuracy using word embeddings and a DNN, which is critical for protecting IoT devices in smart city expansion. |
| S. Malliga. et al. [17] | Looks at denial of service (DoS/DDoS) attacks, focusing on how attack patterns evolve. It examines contemporary deep-learning-based detection algorithms since 2016, classifies attack types, and assesses datasets. The findings indicate the need for improved techniques to dealing with dynamic attacker behavior, noting gaps in the existing literature and recommending future research directions. |
| A. Alzaqebah. et al. [18] | Improves network intrusion detection systems by employing a modified grey wolf optimization algorithm, with a focus on enhanced detection of regular and anomalous traffic. With an accuracy of 81%, an F1 score of 78%, and a G-mean of 84%, the strategy combines filter and wrapper strategies to produce excellent performance, notably in decreasing error rates. The model beats previous meta-heuristic algorithms when tested on the UNSWNB-15 dataset. |
| J. Toldinas. et al. [19] | Describes a novel approach for detecting network intrusions using multistage deep learning image recognition. The suggested method achieves exceptional accuracies of 99.8% for generic attack identification on UNSW-NB15 and 99.7% for DDoS and normal traffic detection on BOUN DDoS by transforming network features into four-channel pictures and leveraging the ResNet50 model. |
| R. Damasevicius. et al. [20] | Introduces LITNET-2020, a novel annotated network benchmark dataset derived from a real-world academic network, addressing the scarcity of realistic datasets for network intrusion detection. With 85 network flow features and 12 attack types, the dataset proves effective in identifying different attack classes, providing a valuable resource for research purposes. |
| M. H. Ali. et al. [21] | Addresses IoT security using a sparse convolutional network for intrusion detection, focusing on DDoS attacks. Trained with intrusion data and characteristics, the network is optimized using evolutionary techniques, effectively minimizing intrusion involvement in IoT data transmission. Experimental results demonstrate superior network security compared with traditional methods. |

As a result, previous researchers investigated a variety of techniques based on standard machine learning for intrusion detection.

## 3. Methods

We learn about the problem and its solutions by studying information and relevant research publications. This knowledge has been translated by us into a framework, which is illustrated in Figure 1 and is divided into three distinct phases. The study makes use of standardized data and is primarily concerned with intrusion detection systems. It specifically makes use of the CSE-CIC-IDS-2018 dataset [22]. Data cleaning, exploratory data analysis,

and data normalization employing two techniques, min–max normalization and Z-score normalization, are all part of the data preprocessing process in Phase 1. This is performed to evaluate the effectiveness of these strategies when applied to different classifier models. Following that, the research moves on to Phase 2, which involves assessing the significance of each feature in the experimental dataset. The goal here is to reduce data complexity, which improves both processing speed and model size. This is accomplished by combining two techniques: principal component analysis (PCA) [23] and random forest (RF) [24]. The investigation includes using the complete dataset without feature reduction, allowing for a comparison of their efficiency when applied to multiple classifier models. In the final phase, the dataset, which has undergone data preprocessing and feature selection based on predetermined criteria, is used with machine learning algorithms chosen through common classification techniques such as XGBoost [25], CART [26], DT [27], KNN [28], MLP [29], RF [30], LR [31], and Bayes [32], which are used to assess performance across multiple dimensions, as mentioned in the following section.

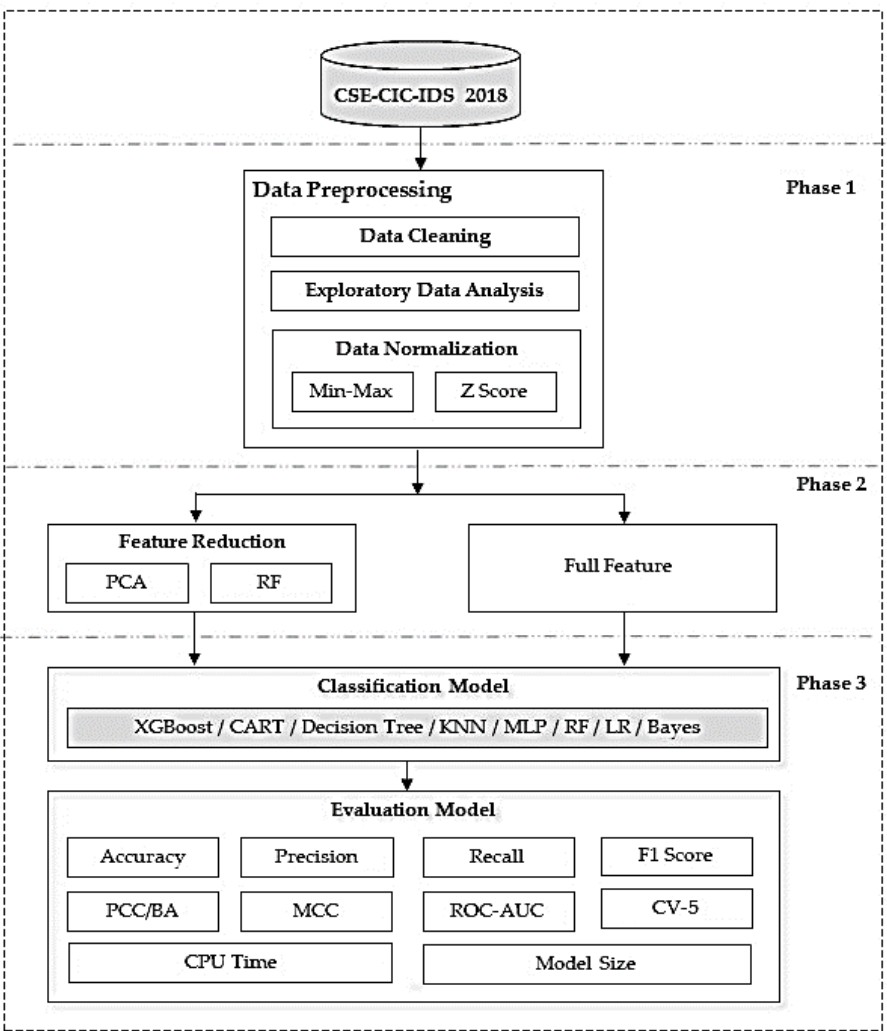

**Figure 1.** The proposed framework.

## 4. Experimental Setup

This study used a 64-bit Windows operating system (Windows 11) with the following specifications: an 11th Gen Intel(R) Core(TM) i7-11800H at 2.30 GHz, 32 GB of 2933 MHz DDR4 memory. Because the amount of large data used in the test is substantial, the Python 3.11 environment was used, with periodic updates that included enhancements such as enhanced language capabilities, faster performance, and the addition of additional libraries

or modules. With each Python release, developers should expect continual improvements in usability, performance, and overall usefulness. For data preparation, handling, preprocessing, analysis, training, and evaluation metrics, the recommended model was built and evaluated using Numpy, Pandas, and Scikit Learn. Pandas and Numpy were used for data handling and preprocessing, while Scikit Learn was used for model training, evaluation, and metrics evaluation. The Seaborn program and Matplotlib were used to visualize the data. Subsequent subsections go into greater detail about the research.

### 4.1. CSE-CIC-IDS-2018 Data Set

The data set given for the CSE-CIC-IDS-2018 [22] was developed through a collaborative project between the Communications Security Establishment (CSE) and the Canadian Institute for Cybersecurity (CIC). It was created with the goal of evaluating intrusion detection research, and it has now become a benchmark dataset for the evaluation of IDSs. This dataset has been meticulously curated and developed to imitate real-world cyber threats and attacks, resulting in a wide and comprehensive set of situations for examination. Its significance arises from its capacity to replicate complicated network environments, allowing academics and practitioners to successfully assess and improve intrusion detection systems. The data was obtained through a ten-day period, eighty columns, and there are fifteen sorts of attacks: FTP-Brute Force, SSH-Brute Force, DoS attacks-GoldenEye, DoS attacks-Slowloris, DoS attacks-Hulk, DoS attacks-SlowHTTPTest, DDoS attacks-LOIC-HTTP, DDOS attack-HOIC, DDOS attack-LOIC-UDP, Brute Force-Web, Brute Force-XSS, SQL injection, infiltration, label, and bot. The study focuses on DDoS intrusions, because of a tough form, difficult to mitigate [14]. From analyzing data over a 10-day period, DDoS intrusions were found on the second day, namely 02-20-2018.csv (84 features) and 02-21-2018.csv (80 features). Therefore, the researcher chose to use this dataset for further investigation.

### 4.2. Data Preprocessing
#### 4.2.1. Data Cleaning

This entailed cutting it down to 80 features by removing the first four: "Flow ID", "Src IP", "Src Port", and "Dst IP". Both datasets, which differ in their feature counts, were chosen for their importance. The dataset was standardized by homogenizing it to 80 features and removing the stated initial properties. The removed attributes from both days were then combined to form a uniform dataset for further investigation. The "Label" column has been converted to numerical values, with Label 0 denoting benign, Label 1 denoting DDoS attacks-LOIC-HTTP, Label 2 denoting DDOS attacks-HOIC, and Label 3 denoting DDOS attacks-LOIC-UDP.

#### 4.2.2. Exploratory Data Analysis

The analysis included determining the minimum, maximum, standard deviation, and mean values of the data for all 80 attributes, including the labels. We removed 10 fields with constant zero values for each instance, including "Bwd PSH Flags", "Fwd URG Flags", "CWE Flag Count", "Fwd Byts/b Avg", "Fwd Pkts/b Avg", "Fwd Blk Rate Avg", "Bwd Pkts/b Avg", and "Bwd Blk Rate Avg". In addition, we removed the "Timestamp" fields to prevent learners from discriminating between attack prediction and attack detection. After deleting the unnecessary features, the dataset is more usable for classification applications. The experiment will yield 8,997,323 rows of data and 69 characteristics. Several processes were required to ensure data quality, including removing "NaN" values (36,767 rows), removing "+inf" and "-inf" values (22,686 rows), and deleting duplicate rows (2,302,927 rows). The dataset was refined to 6,634,943 rows once these cleaning operations were completed, making it appropriate for further research and use. The effect of data cleaning on attack category distribution is shown in Table 2.

**Table 2.** The effect of data cleaning on attack category distribution.

| Type | Original Data | | After Clean Data | |
|---|---|---|---|---|
| Label Feature | Record | Percent | Record | Percent |
| Benign | 7,733,390 | 85.95 | 5,858,988 | 88.31 |
| DoS attacks-LOIC-HTTP | 576,191 | 6.40 | 575,364 | 8.67 |
| DDOS attack-HOIC | 686,012 | 7.62 | 198,861 | 3.00 |
| DDOS attack-LOIC-UDP | 1730 | 0.02 | 1730 | 0.03 |
| total | 8,997,323 | 100.00 | 6,634,943 | 100.00 |

Table 2 displays data statistics before and after cleaning. Initially, there were 8,997,323 rows grouped into different labels, with "Benign" accounting for 85.95% of the records, "DDoS attacks-LOIC-HTTP" accounting for 6.40%, "DDOS attack-HOIC" accounting for 7.62%, and "DDOS attack-LOIC-UDP" accounting for 0.02%. The dataset was cleaned and reduced to 6,634,943 rows. "Benign" entries made up 88.31% of the cleaned data, indicating a reduction from the original dataset. "DDoS attacks-LOIC-HTTP" and "DDOS attack-HOIC" percentages increased somewhat, while "DDOS attack-LOIC-UDP" remained at 0.03%. These modifications represent the effect of data cleansing on the distribution of the various attack categories.

### 4.2.3. Data Normalization

Normalization is used in the data preparation step of machine learning to standardize numerical column values and ensure they are on a consistent scale [33]. Normalization, a transformation method, improves a model's performance and accuracy greatly, especially when the distribution of information is uncertain. Without a consistent pattern, effective normalization relies on large datasets to smooth data by removing outliers. This technique, which is critical in data preprocessing for network intrusion detection systems (NIDS), standardizes data to a given scale, often ranging from 0 to 1. This ensures that all features have consistent scales and ranges, thereby improving the performance and accuracy of NIDS. Several normalization approaches are employed in data pre-processing. Some of the most common are shown in [34].

The decision between min–max scaling and Z-score normalization in the study of the CSE-CIC-IDS-2018 dataset is determined by the peculiarities of the cybersecurity data. Min–max scaling may be useful in retaining the interpretability of feature values, especially when precise ranges are important in the context of network intrusion detection. This approach may be appropriate for checking that normalized features keep their associations and stay within expected limitations. However, given the nature of cybersecurity datasets, Z-score normalization may offer advantages due to the existence of outliers. Its resistance to extreme values may be critical for improving the resilience of intrusion detection models to aberrant network activity. Furthermore, Z-score normalization may contribute to a more effective comparison of varied aspects inside the CSE-CIC-IDS-2018 dataset, thereby increasing the effectiveness of machine learning algorithms built for cybersecurity applications. Finally, empirical evaluations should be used to decide which normalization technique best aligns with the specific characteristics and aims of the intrusion detection task utilizing this dataset.

- Min–Max Normalization: This approach reduces the values of a feature to a range between 0 and 1. It accomplishes this by subtracting the minimum value of the feature from each data point and then dividing the result by the range of the feature. This technique's equivalent mathematical equation is shown as (1), where $X$ is an original value and $X'$ is the normalized value [35]:

$$X' = \frac{(X - X_{min})}{(X_{max} - X_{min})} \tag{1}$$

- Z-score Normalization: This method scales a feature's values to have a mean of 0 and a standard deviation of 1. This is accomplished by removing the feature's mean from each value and then dividing by the standard deviation. Mathematical equation for this strategy is given as (2), where $X$ is an original value and $X'$ is the normalized value [36]:

$$X' = \frac{(X - mean)}{std} \tag{2}$$

### 4.3. Feature Selection

We compared two feature selection methods in this study: principal component analysis (PCA) and fandom forest (RF). The following are the comparison's specifics.

#### 4.3.1. PCA

Principal component analysis is a sophisticated statistical approach used in data analysis and machine learning to reduce complex datasets. Its major goal is to decrease the amount of characteristics or dimensions in a dataset while retaining critical information. PCA does this by changing the original variables into a new set of variables known as principle components. These components, which are linear combinations of the original features, are intentionally made uncorrelated in order to capture the maximum variation in the data. PCA allows academics and data scientists to analyze high-dimensional data more effectively, identify patterns, and maximize the performance of machine learning algorithms by selecting the principal components that elucidate the most variability. PCA, in essence, simplifies both data interpretation and processing by condensing the information into a more comprehensible and insightful format [23].

#### 4.3.2. RF

Random forest, in addition to being a powerful prediction model, is also a useful tool for feature selection in machine learning. Random forest evaluates the value of each feature throughout the training process by determining how much it contributes to lowering impurity or inaccuracy in the model. Higher significance scores are ascribed to features that play a substantial influence in decision making across multiple trees. Data scientists can find the most influential aspects in their dataset by examining these ratings. This inbuilt feature ranking capability simplifies the selection process, allowing practitioners to focus on the factors that will have the greatest impact on their study. The capacity of random forest to perform feature selection improves model efficiency, reduces overfitting, and improves the general interpretability of machine learning systems [24].

### 4.4. Classification Model

Classification predicts data classes, and, in the context of an intrusion detection system (IDS), attacks are categorized as binary or multiclass to discern benign or malicious network traffic. Binary classification involves two classes, while multiclass datasets can have n classes. This complexity imposes a strain on algorithms in terms of computational power and time, perhaps resulting in less effective algorithm outcomes. In the process of classification, each dataset is evaluated and categorized as either typical or unusual. Existing structures are maintained, and new instances are generated. Classification is employed for both identifying irregular patterns and detecting anomalies, although it is more frequently utilized for recognizing misuse. In the current study, eight machine learning techniques were applied, along with feature selection methods addressing class imbalances [37].

#### 4.4.1. XGBoost

XGBoost is a very effective machine learning method noted for its high predicted accuracy and speed. It is classified as ensemble learning since it combines predictions from

numerous decision trees to generate strong models. What distinguishes XGBoost is its emphasis on overcoming the constraints of existing gradient boosting methods, resulting in a highly efficient algorithm. It accomplishes this by training simple models iteratively to repair faults and optimize performance using techniques such as regularization and parallelization. The capacity of XGBoost to handle complicated data relationships has made it a popular choice in a variety of industries, winning multiple machine learning competitions and finding applications in data science and finance [25].

### 4.4.2. CART

CART is a versatile machine learning approach that can solve classification and regression problems. It recursively divides the dataset depending on feature values, yielding a tree structure with each node representing a feature and a split point. This action is repeated until the halting requirements are met, resulting in the formation of a binary tree. CART is well known for its ease of use and interpretability, making it a popular choice in a variety of industries. It is particularly useful for finding non-linear correlations in data and making accurate predictions for both categorical and numerical outcomes [26].

### 4.4.3. DT

A decision tree is a basic machine learning approach used for classification and regression. It divides the dataset recursively into subsets based on the values of the input features. These divisions are determined by choosing qualities and criteria that result in the best class separation or most accurate predictions. Each internal node represents a feature and a split point, and each leaf node represents the output, which is commonly a class label for classification tasks or a numerical value for regression tasks. The method divides the data until a stopping criterion, such as a maximum tree depth or a minimum number of samples at a leaf node, is fulfilled. Because they are simple to read and illustrate, decision trees are popular for exploratory analysis and decision-making processes [27].

### 4.4.4. KNN

KNN is a basic powerful machine learning method that may be used for classification and regression problems. Predictions in KNN are based on the majority class or the average of the k-nearest data points in the feature space. "K" represents the number of nearest neighbors considered, and the method calculates distances between the query point and all other points in the dataset to discover the closest ones. In classification, the most prevalent class among these neighbors determines the forecast, whereas, in regression, the average of the nearby values defines the prediction. KNN is non-parametric and instance-based, which means it makes no assumptions about the underlying data distribution, making it adaptable and simple to grasp. However, its performance can be affected by the option selected [28].

### 4.4.5. Multilayer Perceptron (MLP)

MLP is a machine learning artificial neural network. It is made up of several interconnected layers, including an input layer, one or more hidden layers, and an output layer. Each node connection has a weight, and the network learns by altering these weights during training in order to minimize the discrepancy between expected and actual outputs. MLPs can describe complicated patterns and relationships in data, making them useful for applications like classification, regression, and pattern recognition. They are very good at handling huge and complex datasets because of their capacity to capture nonlinear correlations, but they require careful tuning and a significant amount of training data to avoid overfitting [29].

### 4.4.6. RF

RF is a machine learning technique that, during training, generates a set of decision trees. Each tree in the ensemble is built with a random subset of the data and a random

subset of the features. For regression tasks, the algorithm makes predictions by averaging the forecasts of these individual trees, whereas for classification tasks, the algorithm takes a majority vote. Random forest is well known for its precision, robustness, and ability to handle complex data interactions. It reduces overfitting by pooling the predictions of several trees, making it one of the most popular and powerful machine learning techniques [30]

### 4.4.7. LR

LR is a statistical technique used to perform binary classification tasks. Contrary to its name, it is utilized for classification rather than regression. The algorithm calculates the likelihood that a given input belongs to a specific class. The logistic function (also known as the sigmoid function) is applied to the linear combination of input features and their associated weights. The result is converted into a value between 0 and 1, signifying the likelihood of the input falling into the positive category. If this probability exceeds a certain threshold (typically 0.5), the input is considered positive; otherwise, it is considered negative. Logistic regression is an essential tool in machine learning due to its simplicity, interpretability, and efficiency for linearly separable data [31].

### 4.4.8. Bayes

Naive Bayes is a probabilistic machine learning technique that is used for classification jobs. It is based on Bayes' theorem, which assesses the likelihood of a certain event occurring based on prior knowledge of factors that may be relevant to the occurrence. In the context of naive Bayes, it is assumed that features in the dataset are conditionally independent, which means that the presence of one feature does not affect the presence of another. Despite this simplistic assumption (thus the term "Naive"), naive Bayes performs admirably in many actual applications, particularly text classification and spam filtering. It is computationally efficient, simple to implement, and performs well with huge datasets, making it a popular choice for a variety of classification jobs [32].

### *4.5. Evaluation Model*

This research evaluates an intrusion detection method using nine important criteria: in k-fold cross-validation, accuracy, precision, recall, F1 score, PCC/BA, MCC, ROC, and average were calculated. Classification, CPU time, and model size are also explored.

Evaluation accuracy, sometimes known as accuracy, is a fundamental parameter in analyzing the performance of machine learning models, notably in classification tasks. It computes the proportion of accurately predicted cases out of all instances in the dataset. High accuracy shows that the model's predictions closely match the actual outcomes.

- F1 score contains both recall and precision and the mathematical equation for this strategy is given as (3)

$$\text{F1 Score} = \frac{2 \times (\text{Precision} \times \text{Recall})}{(\text{Precision} + \text{Recall})} \tag{3}$$

  The F1 score provides more weight to the lower of the two values and is the harmonic mean of precision and recall. This indicates that if either precision or recall is low, the F1 score will be much lower as well. However, if both precision and recall are strong, the F1 score will be close to 1. This can result in a biased outcome if one of the measurements is significantly greater than the other [4].

- The Matthews correlation coefficient (MCC) is a more reliable statistical rate that produces a high score only if the prediction performed well in all four confusion matrix categories (true positives, false negatives, true negatives, and false positives), proportionally to the size of positive and negative elements in the dataset. MCC's formula takes into account all of the cells in the confusion matrix. In machine learning, the MCC is used to assess the quality of binary (2-class) classification. MCC is a correlation coefficient that exists between the exact and projected binary classifications and typically returns a value of 0 or 1. mathematical equation for this strategy is given

as (4) [38], where TP as correctly predicted positives are called true positives, FN as wrongly predicted negatives are called false negatives, TN actual negatives that are correctly predicted negatives are called true negatives, and FP actual negatives that are wrongly predicted positives are called false positives:

$$\text{MCC} = \frac{\text{TP} \times \text{TN} - \text{FP} \times \text{FN}}{\sqrt{(\text{TP} + \text{FP})(\text{TP} + \text{FN})(\text{TN} + \text{FP})(\text{TN} + \text{FN})}} \tag{4}$$

- Receiver operating characteristic (ROC) as most indicators can be influenced by dataset class imbalance, making it difficult to rely on a single indication for model differentiation [39]. ROC curves are used to differentiate between attack and benign instances, with the x-axis representing the false alarm rate (FAR) and the y-axis representing the detection rate (DR).
- The probability of correct classification (PCC) is a probability value between 0 and 1 that examines the classifier's ability to detect certain classes. It is critical to understand that relying only on overall accuracy across positive and negative examples might be misleading. Even if our training data is balanced, performance disparities in different production batches are possible. As a result, accuracy alone is not a reliable measure, emphasizing the need of metrics such as PCC, which focus on the classifier's accurate classification probabilities for individual classes.
- Balanced accuracy (BA) is calculated as the average of sensitivity and specificity, or the average of the proportion corrects of each individually. It entails categorizing the data into two categories. The mathematical equation for this strategy is given as (5). When all classes are balanced, so that each class has the same TN number of samples, TP + FN TN + FP and binary classifier's "regular" accuracy is approximately equivalent to balanced accuracy:

$$\text{BA} = 0.5 \times \left( \left( \frac{\text{TP}}{(\text{TP} + \text{FN})} \right) + \left( \frac{\text{TN}}{(\text{TN} + \text{FP})} \right) \right) \tag{5}$$

- ROC score handled the case of a few negative labels similar to the case of a few positive labels. It is worth noting that the F1 score for the model is nearly the same because positive labels are plentiful, and it only cares about positive label misclassification. The probabilistic explanation of the ROC score is that a positive example and a negative case are chosen at random. In this case, rank is defined by the order of projected values.
- Cross-validation (CV) is a statistic used to evaluate the performance of a machine learning model. The dataset is partitioned into k subsets or folds in k-fold cross-validation. The model is trained on one of these folds while being validated on the others. This procedure is performed k times, with each fold only serving as validation data once. To measure total accuracy, the accuracy ratings acquired from each fold are averaged. This method ensures that the model is evaluated over numerous data subsets, reducing the danger of overfitting and producing a more realistic estimate of its performance on unseen data.
- In the context of evaluation, CPU time refers to the overall length of time it takes a CPU to complete a certain job or process. When analyzing algorithms or models, CPU time is critical for determining computational efficiency. Evaluating CPU time helps determine how quickly a given algorithm or model processes data, making it useful for optimizing performance, particularly in applications where quick processing is required, such as real-time systems or large-scale data processing jobs. Lower CPU time indicates faster processing and is frequently used to determine the efficiency and practical applicability of algorithms or models.
- The memory space occupied by a machine learning model when deployed for prediction tasks is referred to as model size in classification. Model size must be considered, especially in applications with limited storage capacity, such as mobile devices or edge computing environments. A lower model size is helpful since it minimizes memory

requirements, allowing for faster loading times and more efficient resource utilization. However, it is critical to strike a balance between model size and forecast accuracy; highly compressed models may forfeit accuracy. As a result, analyzing model size assures that the deployed classification system is not only accurate but also suited for the given computer environment, hence increasing its practicality and usability.

Operating the receiver characteristic values and CPU runtime provides complimentary information on several aspects of machine learning model performance. The area under the ROC curve (AUC-ROC) and other ROC metrics provides a more sophisticated view of a model's discriminatory capacity. This statistic is useful when the balance of precision and recall is critical. CPU runtime, on the other hand, is a practical statistic that addresses a model's computational efficiency, which is critical for real-time applications. It estimates the time required for the model to create predictions, determining deployment feasibility in time-critical applications. ROC values and CPU runtime provide a more thorough evaluation and operational efficiency in the deployment of machine learning models when compared with accuracy, which may not capture class imbalances or computational efficiency. As a result, they are better suited for cases where the data is uneven.

## 5. Experimental Results and Discussions

In Phase 1, we performed preprocessing with data cleaning, exploratory data analysis, and normalization. We double checked for duplicates after selecting features. The dataset is divided into three sections: training, testing, and validation. To begin, the sample data is divided into two parts: 80 percent train data and 20 percent test data (see Figure 2).

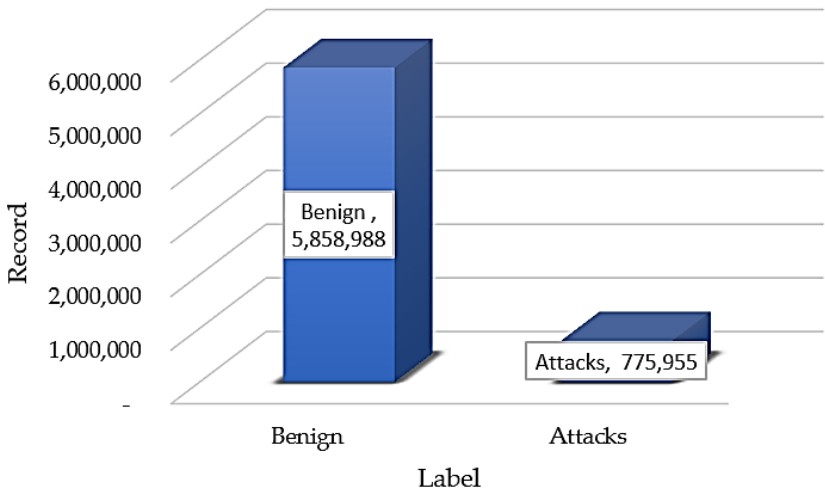

**Figure 2.** Network traffic distribution.

After data cleaning and exploratory data analysis, we normalized the dataset and converted the values of each feature to a specified scale, often ranging from 0 to 1. Min–max normalization is a common method for this purpose, in which data are adjusted to fit inside a given range by subtracting the minimum value and dividing by the range. Z-score normalization is another strategy that standardizes features by subtracting the mean and dividing by the standard deviation, resulting in a mean of 0 and a standard deviation of 1. Normalization is especially crucial for algorithms that are sensitive to varied feature scales, since it ensures constant and fair comparisons of different qualities during the training phase.

In Phase 2, we split the process into two parts. Firstly, they reduced the number of features using PCA and RF techniques, and then fed the processed data into classification models. Secondly, they used all the data without feature reduction and applied various classification models to evaluate the outcomes of data classification, including CPU runtime and model size.

We used PCA to minimize the number of features depending on certain variance ratios, resulting in several feature sets: 11 features as PCA11 for variance ratios greater than or equal to 0.006586494, 9 features as PCA9 for variance ratios greater than or equal to 0.017037139, 7 features as PCA7 for variance ratios greater than or equal to 0.036543147, 5 features as PCA5 for variance ratios greater than or equal to 0.052597381, and 3 features as PCA3 for variance ratios greater than or equal to 0.125926325. Figure 3 depicts the importance of these variance ratios. Once these critical qualities were found, they were employed in Phase 3 for data classification and further evaluation.

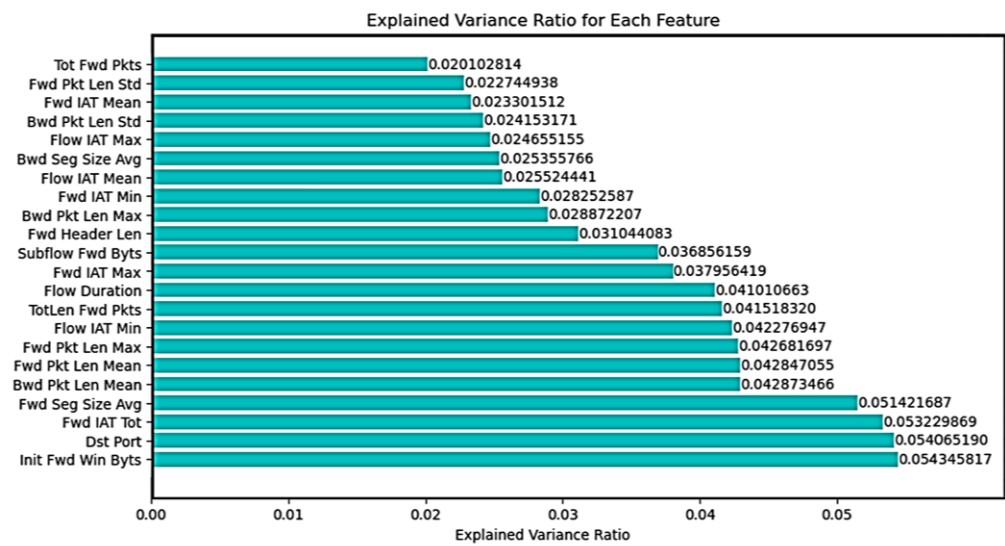

**Figure 3.** Feature selection by using PCA considering variance ratios.

We used random forest (RF) to narrow down the feature set based on particular variance ratios. The following criteria were used to choose the features: 22 features as RF22 for variance ratios greater than or equal to 0.02, 13 features as RF13 for variance ratios greater than or equal to 0.03, and 4 features as RF4 for variance ratios greater than or equal to 0.05 (Figure 4). Following the identification of these essential features, they were employed in Phase 3 for data classification and further evaluation.

**Figure 4.** Feature selection by using RF considering variance ratios.

Phase 3 is the final stage in which the produced dataset is analyzed further. Methods for preprocessing and feature selection are translated into machine learning approaches that are extensively used by researchers in intrusion detection systems. Popular classification algorithms such as XGBoost, CART, DT, KNN, MLP, RF, LR, and Bayes are set up with precise parameter settings, as follows. Key parameters for XGBoost include a learning rate of 0.2, 1000 estimators, a maximum depth of five, and other parameters such as min_child_weight, subsample, and colsample_bytree set to one. CART uses requirements

like squared error, no maximum depth, a minimum sample split of three, and a minimum sample leaf of one. DT and CART have comparable characteristics, although KNN has three neighbors, uniform weights, the "auto" algorithm, and a leaf size of 30. MLP employs (100, 50) hidden layer sizes, 1000 maximum iterations, "relu" activation, and a random state of 42. RF has 40 estimators, 3 maximal features, the "gini" criterion, no maximum depth, and a random state of 42. Logistic regression has a maximum iteration of 8000, the "l2" penalty, fit_intercept set to true, the "lbfgs" solver, and a random state of 42. Finally, the Bayes employs default parameters, with priors set to none and var_smoothing set to $1 \times 10^{-9}$. The evaluation of performance encompasses several dimensions, and the results are summarized here in Table 3.

**Table 3.** Summary of classifier performance metrics using min–max and Z-score normalization.

| Classifiers | Accuracy | Precision | Recall | F1 Score | PCC/BA | MCC | ROC | CV 5 | CPU Time (S) | Model Size (KB) |
|---|---|---|---|---|---|---|---|---|---|---|
| **Min–Max** | | | | | | | | | | |
| XGBoost | 0.999950 | 0.975427 | 0.982578 | 0.978946 | 0.982578 | 0.999765 | 0.991281 | 0.999930 | 92.86 | 590.85 |
| CART | 0.999917 | 0.967775 | 0.960877 | 0.964270 | 0.960877 | 0.999609 | 0.980424 | 0.997995 | 112.79 | 57.31 |
| DT | 0.999911 | 0.958447 | 0.960853 | 0.959643 | 0.960853 | 0.999580 | 0.980413 | 0.999889 | 65.41 | 68.74 |
| RF | 0.999631 | 0.956304 | 0.982593 | 0.968626 | 0.982593 | 0.998260 | 0.991064 | 0.999560 | 131.67 | 7860.20 |
| Bayes | 0.950489 | 0.747992 | 0.984988 | 0.831505 | 0.984988 | 0.820660 | 0.986005 | 0.950566 | 7.02 | 6.65 |
| LR | 0.992956 | 0.898140 | 0.989497 | 0.937082 | 0.989497 | 0.967303 | 0.992159 | 0.989964 | 860.09 | 4.58 |
| MLP | 0.999835 | 0.938870 | 0.993869 | 0.962856 | 0.993869 | 0.999221 | 0.996879 | 0.998905 | 2220.53 | 291.93 |
| KNN | 0.999848 | 0.947640 | 0.963627 | 0.955322 | 0.963627 | 0.999281 | 0.981772 | 0.999815 | 6460.54 | 2,861,321.35 |
| **Z-score** | | | | | | | | | | |
| XGBoost | 0.999948 | 0.977524 | 0.977526 | 0.977525 | 0.977526 | 0.999755 | 0.988754 | 0.999934 | 89.12 | 581.15 |
| CART | 0.999921 | 0.968903 | 0.964489 | 0.966674 | 0.964489 | 0.999626 | 0.982229 | 0.997749 | 151.42 | 56.89 |
| DT | 0.999918 | 0.960671 | 0.967360 | 0.963963 | 0.967360 | 0.999612 | 0.983669 | 0.999881 | 76.50 | 68.93 |
| RF | 0.999739 | 0.966118 | 0.982353 | 0.973918 | 0.982353 | 0.998769 | 0.991008 | 0.999603 | 153.25 | 17,055.20 |
| Bayes | 0.949471 | 0.752121 | 0.984739 | 0.834724 | 0.984739 | 0.817708 | 0.985739 | 0.950838 | 7.43 | 6.65 |
| LR | 0.996893 | 0.912855 | 0.994199 | 0.947252 | 0.994199 | 0.985584 | 0.996643 | 0.995351 | 6920.85 | 4.58 |
| MLP | 0.998974 | 0.923494 | 0.998375 | 0.954336 | 0.998375 | 0.995160 | 0.998676 | 0.998805 | 1167.41 | 291.81 |
| KNN | 0.999840 | 0.945759 | 0.967207 | 0.955923 | 0.967207 | 0.999246 | 0.983554 | 0.999803 | 11,468.02 | 2,861,321.35 |

Table 3 contains two sections: normalized data using the min–max and Z-score. The min–max normalization section presents the performance metrics of various classifiers. XGBoost outperforms in all categories, including accuracy (0.999950), precision (0.975427), recall (0.982578), and F1 score (0.978946). It also has a high MCC and area under the ROC curve, showing that it performs well overall. DT and CART classifiers outperform XGBoost in terms of accuracy and balanced metrics, but with smaller model sizes and cheaper computing costs. RF has a high recall rate (0.982593) but a much greater model size and computational load. The recall of Bayes is impressive (0.984988), but it comes at the sacrifice of precision and overall accuracy. LR achieves an excellent balance of precision and recall, whereas MLP and KNN, respectively, specialize in high precision and recall. The classifier should be chosen based on specific needs such as accuracy, computational efficiency, or the trade-off between precision and recall, while also taking into account aspects such as model size and processing time. The performance metrics of the classifiers based on Z-score scaling are reported in this investigation. XGBoost delivers high accuracy (0.999948) as well as high precision, recall, F1 score, and MCC. DT and CART classifiers outperform XGBoost in a variety of metrics while being more computationally efficient and requiring smaller model sizes. RF has a high recall rate (0.982353), but it has a much greater model size and a higher computational cost. Bayes excels in recall at the expense of precision and overall accuracy. LR achieves a good mix of accuracy and recall, whereas MLP has a high recall and KNN has a high precision. Specific needs, like accuracy, computational efficiency, or trade-offs between precision and recall should be considered when selecting a classifier, as should model size and processing time.

Because of the multiple evaluation criteria available, we chose to consider the ROC values, as well as the CPU time and model size. Among these factors, we chose three

classifiers: DT, XGBoost, and RF, all of which produced very comparable evaluation findings. This choice was made when conducting feature selection trials.

Following that, the model was used in conjunction with feature selection approaches such as PCA and RF. Table 4 displays the results of these tests.

**Table 4.** A comparison of classifier performance with different feature selection and normalization techniques.

| Classifiers | Accuracy | Precision | Recall | F1 Score | PCC/BA | MCC | ROC | CV 5 | CPU Time (S) | Model Size (KB) |
|---|---|---|---|---|---|---|---|---|---|---|
| Min-Max | | | | | | | | | | |
| RF-PCA11 | 0.996154 | 0.925325 | 0.960638 | 0.942236 | 0.960638 | 0.982109 | 0.979327 | 0.997329 | 135.52 | 31,307.31 |
| RF-PCA9 | 0.996145 | 0.926899 | 0.960586 | 0.943059 | 0.960586 | 0.982067 | 0.979291 | 0.997325 | 131.46 | 31,335.76 |
| RF-PCA7 | 0.996159 | 0.927997 | 0.960655 | 0.943677 | 0.960655 | 0.982134 | 0.979337 | 0.997324 | 137.16 | 31,305.15 |
| RF-PCA5 | 0.972420 | 0.869089 | 0.770071 | 0.789002 | 0.770071 | 0.863637 | 0.859193 | 0.991191 | 168.64 | 55,616.56 |
| RF-PCA3 | 0.958392 | 0.832067 | 0.652148 | 0.655230 | 0.652148 | 0.788547 | 0.787589 | 0.977432 | 197.05 | 334,645.57 |
| RF-RF22 | 0.999870 | 0.955205 | 0.975940 | 0.965073 | 0.975940 | 0.999385 | 0.987934 | 0.999761 | 188.84 | 10,281.29 |
| RF-RF13 | 0.999920 | 0.960641 | 0.970969 | 0.965681 | 0.970969 | 0.999623 | 0.985472 | 0.999881 | 154.06 | 5400.85 |
| RF-RF4 | 0.999837 | 0.913762 | 0.983025 | 0.942297 | 0.983025 | 0.999232 | 0.991467 | 0.999819 | 129.30 | 1241.54 |
| DT-PCA11 | 0.996097 | 0.925248 | 0.940889 | 0.932627 | 0.940889 | 0.981836 | 0.969390 | 0.997278 | 8.67 | 1173.92 |
| DT-PCA9 | 0.996098 | 0.925831 | 0.940891 | 0.932914 | 0.940891 | 0.981840 | 0.969392 | 0.997278 | 6.77 | 1174.43 |
| DT-PCA7 | 0.996099 | 0.925985 | 0.941614 | 0.933358 | 0.941614 | 0.981843 | 0.969753 | 0.997283 | 6.20 | 1174.20 |
| DT-PCA5 | 0.971348 | 0.861854 | 0.743431 | 0.769738 | 0.743431 | 0.858021 | 0.844684 | 0.981260 | 5.35 | 2028.59 |
| DT-PCA3 | 0.957864 | 0.829175 | 0.643716 | 0.647520 | 0.643716 | 0.785412 | 0.782492 | 0.959822 | 5.95 | 12,713.26 |
| DT-RF22 | 0.999918 | 0.961201 | 0.963030 | 0.962111 | 0.963030 | 0.999612 | 0.981504 | 0.999884 | 40.10 | 83.07 |
| DT-RF13 | 0.999916 | 0.960222 | 0.964468 | 0.962324 | 0.964468 | 0.999602 | 0.982222 | 0.999882 | 25.03 | 84.21 |
| DT-RF4 | 0.999836 | 0.913430 | 0.983022 | 0.942066 | 0.983022 | 0.999224 | 0.991465 | 0.999816 | 5.47 | 42.21 |
| XGBoost-PCA11 | 0.997706 | 0.920757 | 0.988790 | 0.949388 | 0.988790 | 0.989180 | 0.993236 | 0.997635 | 50.74 | 660.01 |
| XGBoost-PCA9 | 0.997705 | 0.920756 | 0.988784 | 0.949385 | 0.988784 | 0.989177 | 0.993233 | 0.997634 | 46.13 | 667.18 |
| XGBoost-PCA7 | 0.997705 | 0.920752 | 0.988784 | 0.949383 | 0.988784 | 0.989177 | 0.993233 | 0.997631 | 44.52 | 661.26 |
| XGBoost-PCA5 | 0.994354 | 0.902693 | 0.982245 | 0.936947 | 0.982245 | 0.973601 | 0.988806 | 0.994281 | 45.00 | 787.88 |
| XGBoost-PCA3 | 0.984300 | 0.858634 | 0.938125 | 0.893717 | 0.938125 | 0.927370 | 0.961847 | 0.983735 | 44.21 | 741.86 |
| XGBoost-RF22 | 0.999940 | 0.969710 | 0.981110 | 0.975265 | 0.981110 | 0.999719 | 0.990545 | 0.999933 | 54.15 | 572.66 |
| XGBoost-RF13 | 0.999917 | 0.956144 | 0.974560 | 0.964956 | 0.974560 | 0.999609 | 0.987267 | 0.999916 | 44.84 | 573.72 |
| XGBoost-RF4 | 0.999812 | 0.912773 | 0.976517 | 0.939387 | 0.976517 | 0.999111 | 0.988202 | 0.999809 | 40.56 | 564.10 |
| Z-score | | | | | | | | | | |
| RF-PCA11 | 0.997387 | 0.939662 | 0.948200 | 0.943840 | 0.948200 | 0.987658 | 0.972616 | 0.997016 | 139.92 | 40,746.06 |
| RF-PCA9 | 0.997396 | 0.940383 | 0.955569 | 0.947705 | 0.955569 | 0.987658 | 0.976323 | 0.996991 | 145.87 | 40,614.20 |
| RF-PCA7 | 0.997396 | 0.939566 | 0.953297 | 0.946201 | 0.953297 | 0.987702 | 0.975172 | 0.997000 | 143.99 | 40,533.90 |
| RF-PCA5 | 0.991311 | 0.933681 | 0.913607 | 0.922487 | 0.913607 | 0.958517 | 0.949896 | 0.991049 | 183.93 | 69,791.18 |
| RF-PCA3 | 0.962557 | 0.854850 | 0.692485 | 0.724899 | 0.692485 | 0.812001 | 0.813312 | 0.978074 | 222.77 | 353,408.88 |
| RF-RF22 | 0.999882 | 0.958170 | 0.977381 | 0.967350 | 0.977381 | 0.999441 | 0.988660 | 0.999794 | 202.37 | 12,587.54 |
| RF-RF13 | 0.999918 | 0.957692 | 0.977453 | 0.967121 | 0.977453 | 0.999612 | 0.988714 | 0.999879 | 192.22 | 5940.48 |
| RF-RF4 | 0.999837 | 0.913762 | 0.983025 | 0.942297 | 0.983025 | 0.999232 | 0.991467 | 0.999817 | 130.35 | 1221.85 |
| DT-PCA11 | 0.997312 | 0.941784 | 0.940745 | 0.941254 | 0.940745 | 0.987290 | 0.968679 | 0.996832 | 9.58 | 1290.92 |
| DT-PCA9 | 0.997311 | 0.941636 | 0.940022 | 0.940820 | 0.940022 | 0.987286 | 0.968317 | 0.996831 | 7.39 | 1289.56 |
| DT-PCA7 | 0.997311 | 0.942492 | 0.938582 | 0.940528 | 0.938582 | 0.987286 | 0.967597 | 0.996835 | 6.65 | 1289.70 |
| DT-PCA5 | 0.990910 | 0.931768 | 0.896305 | 0.912423 | 0.896305 | 0.956527 | 0.940553 | 0.990634 | 5.99 | 2385.40 |
| DT-PCA3 | 0.958382 | 0.839611 | 0.655594 | 0.670713 | 0.655594 | 0.788676 | 0.789885 | 0.977744 | 6.86 | 13,686.38 |
| DT-RF22 | 0.999913 | 0.959644 | 0.960858 | 0.960249 | 0.960858 | 0.999591 | 0.980417 | 0.999889 | 45.22 | 84.20 |
| DT-RF13 | 0.999916 | 0.959630 | 0.964470 | 0.962023 | 0.964470 | 0.999605 | 0.982224 | 0.999882 | 29.13 | 82.90 |
| DT-RF4 | 0.999836 | 0.913430 | 0.983022 | 0.942066 | 0.983022 | 0.999224 | 0.991465 | 0.999816 | 6.44 | 41.84 |
| XGBoost-PCA11 | 0.997698 | 0.920740 | 0.988776 | 0.949373 | 0.988776 | 0.989142 | 0.993227 | 0.997635 | 50.89 | 667.12 |
| XGBoost-PCA9 | 0.997693 | 0.920721 | 0.988770 | 0.949360 | 0.988770 | 0.989117 | 0.993225 | 0.997634 | 45.37 | 668.91 |
| XGBoost-PCA7 | 0.997697 | 0.920736 | 0.988770 | 0.949367 | 0.988770 | 0.989138 | 0.993223 | 0.997633 | 42.65 | 673.14 |
| XGBoost-PCA5 | 0.994341 | 0.902696 | 0.982201 | 0.936926 | 0.982201 | 0.973534 | 0.988765 | 0.994289 | 42.59 | 784.36 |
| XGBoost-PCA3 | 0.984288 | 0.858767 | 0.939827 | 0.894489 | 0.939827 | 0.927392 | 0.962789 | 0.983736 | 42.28 | 732.69 |
| XGBoost-RF22 | 0.999943 | 0.968946 | 0.984719 | 0.976556 | 0.984719 | 0.999733 | 0.992350 | 0.999935 | 52.72 | 573.66 |
| XGBoost-RF13 | 0.999918 | 0.956262 | 0.975282 | 0.965350 | 0.975282 | 0.999612 | 0.987628 | 0.999916 | 44.31 | 573.12 |
| XGBoost-RF4 | 0.999812 | 0.912765 | 0.976521 | 0.939385 | 0.976521 | 0.999111 | 0.988207 | 0.999809 | 40.07 | 563.23 |

Table 4 compares the efficacy of DT, XGBoost, and RF classifiers. The comparison is based on the use of both the min–max and Z-score normalization methods, as well as the

deployment of feature selection approaches. The major goal is to shorten CPU runtime and reduce model size. It can be explained as follows.

In the min–max normalization and feature selection with PCA and RF section, the data presented provides a full comparison of various classifier settings as well as their performance indicators. Higher PCA dimensions often lead to greater accuracy, precision, and recall when assessing RF models with various feature selection approaches (PCA) and dimensions. Notably, RF-PCA11 and RF-PCA9 have accuracy levels more than 0.996145, illustrating the efficiency of feature selection in improving model performance. DT models combined with PCA also provide competitive accuracy, particularly at higher PCA dimensions. When the RF and XGBoost models are coupled, they exhibit extraordinary precision and recall, making them strong options for applications requiring balanced performance. When determining the best configuration for a given task, it is critical to consider the trade-offs between accuracy, computational complexity (as measured by CPU time), and model size. This study emphasizes the significance of carefully selecting feature selection strategies and classifier combinations to produce best results tailored to individual needs. To improve understanding of model performance evaluation metrics, the researcher showed the data in the form of a radar graph, as shown in Figure 5.

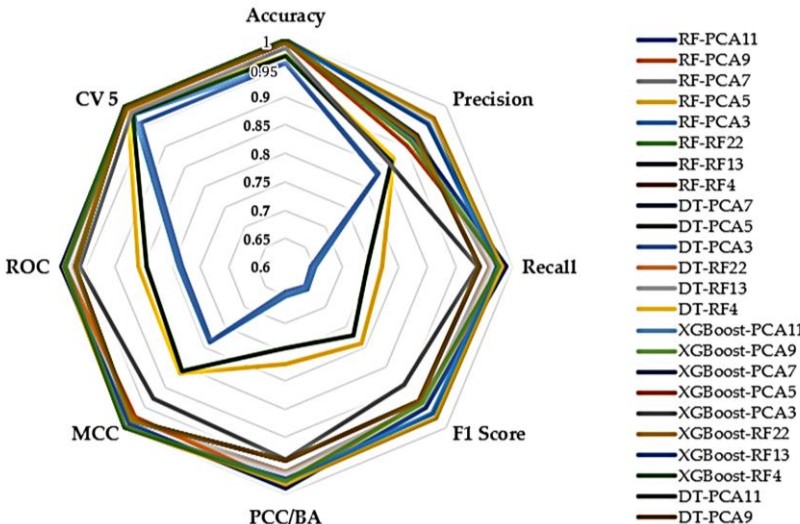

**Figure 5.** Radar chart for classification performance with feature selection and min–max normalization.

The data supplied demonstrates a thorough evaluation of multiple classifiers' performance measures using Z-score scaling. When examining RF models in conjunction with principal component analysis (PCA) at various dimensions, greater PCA dimensions typically result in improved accuracy, precision, and recall. Specifically, RF-PCA11 and RF-PCA9 exhibit outstanding accuracy above 0.997387, demonstrating PCA's usefulness in optimizing model outputs. DT models paired with PCA also perform well, especially with larger PCA dimensions. Furthermore, combining RF and XGBoost with PCA results in good precision and recall, making them solid candidates for applications requiring balanced performance. However, when choosing the optimal model configuration, it is critical to carefully analyze the trade-offs between accuracy and computational complexity, as indicated by CPU time and model size. This analysis emphasizes the importance of choosing appropriate PCA dimensions and classifier combinations to produce optimal and personalized outcomes based on unique job requirements. To improve the understanding of model performance evaluation metrics, the researcher showed the data in the form of a radar graph, as shown in Figure 6.

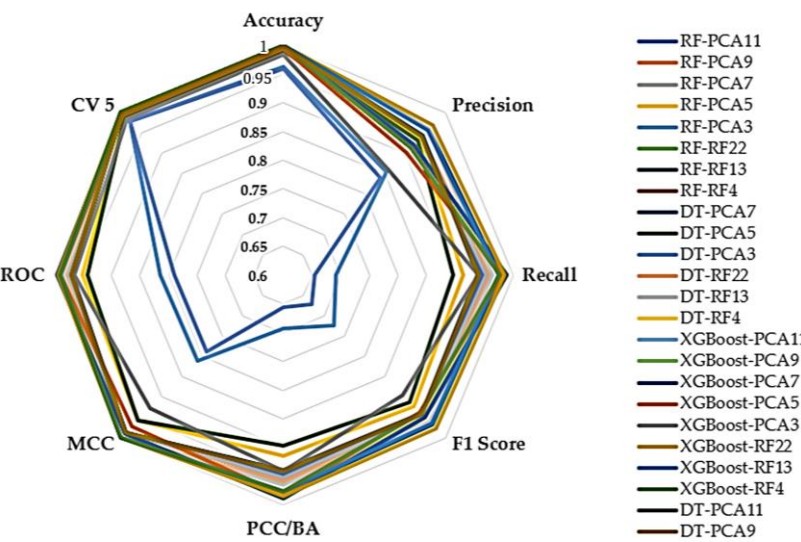

**Figure 6.** Radar chart for classification performance with feature selection and Z-score normalization.

Following that, the models were used in conjunction with feature selection approaches such as PCA and RF. When used in conjunction with XGBoost, feature selection using PCA employing 11 features produced the best performance (considering ROC values combined with CPU run time). This was true whether the data was standardized using min–max or Z-score approaches, because the evaluation findings and CPU processing times were extremely similar (insignificant differences). As a result, both methodologies can be used effectively. Table 5 displays the PCA features that were chosen, a total of 11 variables.

**Table 5.** List of features and importance scores with PCA11.

| Feature Name | Importance Score |
| --- | --- |
| Dst Port | 0.391406787 |
| Protocol | 0.170200700 |
| Flow Duration | 0.125926325 |
| Tot Fwd Pkts | 0.076885293 |
| Tot Bwd Pkts | 0.052597381 |
| TotLen Fwd Pkts | 0.042883536 |
| TotLen Bwd Pkts | 0.037560629 |
| Fwd Pkt Len Max | 0.036543147 |
| Fwd Pkt Len Min | 0.017037139 |
| Fwd Pkt Len Mean | 0.014007651 |
| Fwd Pkt Len Std | 0.012034106 |

Compares various models to the CSE-CIC-IDS-2018 dataset, measuring their accuracy, training time, and other performance measures. S. Ullah et al. [14] employed a decision tree (DT) with random feature selection (30 features) to achieve an astounding 0.9998 accuracy in a very low training period (0.18 s). M. A. Khan. [15] used random feature selection to implement an HCRNNIDS model, obtaining 0.9775 accuracy in 200–250 s. F1 score and precision values were not specified. J. Kim. et al. [10] used a convolutional neural network (CNN) with manual feature extraction. In a training duration ranging from 300 to 900 s, we achieved an accuracy of 0.960. F1 score, precision, and recall measures were not provided in detail. R. Qusyairi. et al. [3] applied an ensemble model with 23 randomly chosen features. Although the accuracy was 0.988, no precise F1 score, precision, or recall statistics were provided. S. Chimphlee. et al. [4] used min–max normalization, random forest feature selection, and class balance (SMOTE), as well as multi-layer perceptron (MLP). A high accuracy of 0.99462 was achieved, with significant precision and recall values.

In our Model 1 and Model 2, both models used XGBoost with principal component analysis (PCA) to pick features. Our Model 1 was 0.997706 accurate, while our Model 2

was 0.997698 accurate. Both models performed well across multiple parameters, including F1 score, precision, recall, ROC, and MCC. Our findings differ from those of S. Ullah. et al. [14], particularly in terms of CPU runtime and accuracy. They applied the SMOTE balance strategy, which resulted in a smaller dataset that was only focused on CPU runtime during training in a binary class environment. In contrast, we used the entire dataset for training and testing without using separate balance methods, multiple classification classes, or reporting combined CPU runtimes. These methodological variations are most likely responsible for the disparities in CPU runtime and accuracy results between our study and theirs. In conclusion, the proposed models demonstrate a variety of approaches, with PCA being particularly helpful in lowering feature dimensions while maintaining high accuracy. The models yield remarkable results in intrusion detection, reflecting the ongoing progress in the field of machine learning applied to cybersecurity. A comparison of the results of the intrusion detection model employing dataset CSE-CIC-IDS-2018 is shown in Table 6.

**Table 6.** Comparison of intrusion detection models using the CSE-CIC-IDS-2018 dataset.

| Study | Method | Feature Selection | CPU Time (s) | Accuracy | F1 Score | Precision | Recall | ROC | MCC | PCC/BA |
|---|---|---|---|---|---|---|---|---|---|---|
| S. Ullah. et al. [14] | DT Class Balance (SMOTE) | RF (30 features) | 0.18 (Train Time) | 0.9998 | n/a | n/a | n/a | n/a | n/a | n/a |
| M. A. Khan. [15] | HCRNNIDS | RF | 200–250 (Train Time) | 0.9775 | 0.976 | n/a | n/a | n/a | n/a | n/a |
| J. Kim. et al. [10] | CNN | Manual Feature Extraction | 300–900 (Train Time) | 0.960 | n/a | n/a | n/a | n/a | n/a | n/a |
| R. Qusyairi. et al. [3] | Ensemble Model | Chi-Square and Spearman's Rank (23 Features) | n/a | 0.988 | 0.979 | n/a | n/a | n/a | n/a | n/a |
| S. Chimphlee. et al. [4] | MLP (Min–Max Normalization, Class Balance (SMOTE) | RF (16 Features) | n/a | n/a | 0.99462 | n/a | n/a | 0.99311 | 0.98151 | 0.99334 |
| Our Model 1 | XGBoost (Min–Max Normalization) | PCA (11 Features) | 50.09 (All Time) Train and Test Time | 0.997706 | 0.949388 | 0.920757 | 0.98879 | 0.993236 | 0.989180 | 0.98879 |
| Our Model 2 | XGBoost (Z-score Normalization) | PCA (11 Features) | 50.89 (All Time) Train and Test Time | 0.997698 | 0.949373 | 0.92074 | 0.988776 | 0.993227 | 0.989142 | 0.988776 |

## 6. Conclusions

After examining the data, it was discovered that three models, namely XGBoost, DT, and RF, had remarkable performance in terms of both ROC values and CPU runtime. As a result, these models were evaluated further in conjunction with feature selection techniques combining PCA and RF. Finally, the combination of XGBoost for classification and feature selection with PCA, resulting in 11 features, produced the best ROC and CPU runtime values. Interestingly, the usefulness of these values remained consistent regardless of whether normalization approaches such as min–max or Z-score were used; the changes seen were not significant. Machine learning classification techniques, as is widely accepted, can be used to assess and anticipate infiltrations. The algorithm performed admirably after preprocessing tactics and feature selection approaches were applied. Although this strategy outperformed others, its utility may be limited in some cases. We argue that the trained models are not yet ready for use in real-world scenarios. Existing models must be improved, and new algorithms developed to address the issues given by unbalanced datasets.

Despite their success in tackling infiltration prediction tasks after preprocessing and feature selection, these models' applicability in specific contexts may be limited. The study recognizes the need for future enhancements to existing models, particularly to address

the issues given by unbalanced datasets. Furthermore, the lack of dynamic testing with multiple types of infiltration raises concerns about the models' adaptability to diverse and changing real-world settings. While the method performed admirably under controlled conditions, its application in real-world settings necessitates refining and thorough testing to assure robustness and broader applicability.

In the future, we intend to examine the impact of deep learning on increasing time complexity and model size, consequently boosting the efficiency of network intrusion detection in real-time settings. Furthermore, the study broadens its scope to handle dataset balancing issues and investigates deployment utilizing the LITNET-2020 and BOUN DDoS datasets.

**Author Contributions:** S.S.: conceptualization, methodology, data curation, software, writing—original draft preparation, writing—review and editing, investigation, validation, and visualization; T.S.: supervision, formal analysis, resources, writing—review and editing, validation, and investigation; T.P.: writing original draft, writing review and editing, validation, and visualization. All authors have read and agreed to the published version of the manuscript.

**Funding:** This research received no external funding.

**Data Availability Statement:** The CSE-CIC-IDS-2018 dataset used in the research is available at the following link: https://registry.opendata.aws/cse-cic-ids2018/ accessed on 1 May 2023.

**Conflicts of Interest:** The authors declare no conflict of interest.

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
