# Peer review of "Optimizing Intrusion Detection Systems in Three Phases on the CSE-CIC-IDS-2018 Dataset"

_computers, doi:10.3390/computers12120245_

Round 1
Reviewer 1 Report
Comments and Suggestions for Authors
The manuscript demonstrates a fundamental understanding of the subject but falls short in terms of methodological rigor and depth of analysis. The study lacks a clear narrative and fails to convincingly argue its case or establish its significance in the field. Given the significant issues in methodology, analysis, and presentation, I recommend major revisions. With substantial revisions and a more rigorous approach, the manuscript could be reconsidered for publication.
My detailed comments are presented below:
1. The title is somewhat convoluted and could be streamlined for better clarity.
2. The abstract provides a general overview but lacks specific insights into the key findings and their implications.
3. The introduction fails to adequately establish the context and significance of the research within the broader field of intrusion detection systems. I suggest to add an overview of the recent network intrusion datasets such as LITNET-2020, and summarize as a table. Additional topics could be discussed extensively such as Decision Tree with Pearson Correlation-based Recursive Feature Elimination Model for Attack Detection in IoT Environment; A Comprehensive Review of Deep Learning Techniques for the Detection of (Distributed) Denial of Service Attacks; A Modified Grey Wolf Optimization Algorithm for an Intrusion Detection System; Threat Analysis and Distributed Denial of Service (DDoS) Attack Recognition in the Internet of Things (IoT); A novel approach for network intrusion detection using multistage deep learning image recognition.
4. There is a lack of a clear, concise statement of the research problem and how this study aims to address it.
5. The hardware and software specifications are detailed, but there is no justification for why these particular tools and environment were chosen. The choice of Python 3.11 and specific libraries should be rationalized in terms of their relevance and advantages for this research.
6. While the CSE-CIC-IDS-2018 dataset is described, there is insufficient discussion on why this dataset is appropriate for the study’s objectives.
7. The selection of specific days' data from the dataset seems arbitrary and lacks a clear rationale.
8. The data cleaning process is outlined, but the reasons for removing certain features are not adequately explained.
9. The choice of normalization techniques (min-max and z-score) is not justified in the context of the dataset and the specific challenges it presents.
10. It is not clear how you deal with dataset imbalance. Balancing the dataset may have improved the performance.
11. There is a lack of detail on the model development process, particularly in the selection and tuning of machine learning algorithms.
12. The evaluation metrics are mentioned, but there is no discussion on why these particular metrics were chosen and how they effectively measure the performance of the models.
13. The presentation of results is somewhat disorganized and lacks a clear structure.
14. There is an overemphasis on the performance of certain models without a balanced discussion of their limitations or potential biases.
15. The discussion does not sufficiently compare the study's findings with existing literature or similar studies. There is a lack of critical analysis regarding the practical applicability of the findings, especially in real-world intrusion detection scenarios.
16. The conclusion is overly optimistic and does not adequately reflect the limitations and scope of the study. Future research directions are mentioned, but they are vague and lack specificity.
Author Response
Dear Reviewer 1
I aim to present the results of the improvements made, incorporating the suggestions provided in the aforementioned revisions. Throughout this process, careful consideration has been given to the proposals made by another reviewer, resulting in a collaborative effort to refine and enhance the content. The consolidated outcomes are reflected in the table presented below:
Author's Reply to the Review Report (Reviewer 1)
Comments and Suggestions for Authors |
Operation |
1.The title is somewhat convoluted and could be streamlined for better clarity. |
Reviewed and revised the article title based on the reviewer's suggestions to simplify it from the original title: "Optimizing Intrusion Detection Systems: Exploring Feature Selection, Normalization, and Three-Phase Precision on the CSE-CIC-IDS-2018 Dataset." The updated title is now " Optimizing Intrusion Detection Systems in Three-Phases on the CSE-CIC-IDS-2018 Dataset". (Lines 1-3) |
2. The abstract provides a general overview but lacks specific insights into the key findings and their implications |
Modifications have been made according to the first reviewer's suggestion, by reviewing and adjusting the description in the abstract according to the second reviewer's suggestion at the same time. Adjustments were made in accordance with the recommendations. Lines 14-23 are from the original article. The revised article can be seen in Lines 13-22 of the new version. |
|
|
3. The introduction fails to adequately establish the context and significance of the research within the broader field of intrusion detection systems. I suggest to add an overview of the recent network intrusion datasets such as LITNET-2020, and summarize as a table. Additional topics could be discussed extensively such as Decision Tree with Pearson Correlation-based Recursive Feature Elimination Model for Attack Detection in IoT Environment; A Comprehensive Review of Deep Learning Techniques for the Detection of (Distributed) Denial of Service Attacks; A Modified Grey Wolf Optimization Algorithm for an Intrusion Detection System; Threat Analysis and Distributed Denial of Service (DDoS) Attack Recognition in the Internet of Things (IoT); A novel approach for network intrusion detection using multistage deep learning image recognition. |
Additional enhancements have been added in response to the reviewer's comments, and a summary table is provided below for convenient reference. This detailed description is mentioned on lines 133-134 of Section 2 Related Work (Table 1).The rewritten article includes the enhanced material and reflects the collaborative efforts made to answer and incorporate the reviewer's important ideas. |
4. There is a lack of a clear, concise statement of the research problem and how this study aims to address it. |
Refinements have been made in response to the suggestions made. Additional details have been introduced in Section 1 Introduction (Lines 65-76) to improve the overall content and address the feedback received. |
5. The hardware and software specifications are detailed, but there is no justification for why these particular tools and environment were chosen. The choice of Python 3.11 and specific libraries should be rationalized in terms of their relevance and advantages for this research. |
We have added a lengthy explanation in Section 4 Experimental Setup (Lines 160-172) to the updated version of the article clarifying the rationale behind the simultaneous use of both hardware and software components in the Experimental Setup. This feature seeks to provide a clear grasp of the benefits of integrating hardware and software. |
6. While the CSE-CIC-IDS-2018 dataset is described, there is insufficient discussion on why this dataset is appropriate for the study’s objectives. |
Corrections have been made to the amended article by including a clear explanation (Lines 174-182) of the relevance of the chosen CSE-CIC-IDS-2018 dataset within Section 4.1 |
7. The selection of specific days' data from the dataset seems arbitrary and lacks a clear rationale. |
In the revised article, enhancements have been implemented by incorporating a lucid explanation in Section 4.1 (Lines 186-190) detailing the rationale behind selecting data from two specific days out of the ten-day period. |
8.The data cleaning process is outlined, but the reasons for removing certain features are not adequately explained. |
Make enhancements as advised By including an explanation in section 4.2.2 Exploratory Data Analysis (Line 207-218) |
9. The choice of normalization techniques (min-max and z-score) is not justified in the context of the dataset and the specific challenges it presents. |
Make enhancements as advised By including an explanation in section 4.2.3 Data Normalization (Line 248-261) |
10. It is not clear how you deal with dataset imbalance. Balancing the dataset may have improved the performance. |
In this research work, the decision was made to use the complete dataset, taking an inclusive approach without truncation. The goal is to cover all of the data. Following research initiatives will move the focus to correcting dataset imbalances as a targeted topic of exploration. |
11. There is a lack of detail on the model development process, particularly in the selection and tuning of machine learning algorithms. |
Enhancements have been made in response to the reviewer's suggestions. Continue by explaining the necessary parameter values for each categorization algorithm, As “Popular classification algorithms such as XGBoost, CART, DT, KNN, MLP, RF, LR, and Bayes are setup with precise parameter settings as follows. Key parameters for XGBoost include a learning rate of 0.2, 1000 estimators, a maximum depth of 5, and other parameters such as min_child_weight, subsample, and colsample_bytree set to 1. CART uses requirements like squared error, no maximum depth, a minimum sample split of three, and a minimum sample leaf of one. DT and CART have comparable characteristics, although KNN has 3 neighbors, uniform weights, the 'auto' algorithm, and a leaf size of 30. MLP employs (100, 50) hidden layer sizes, 1000 maximum itera-tions, 'relu' activation, and a random state of 42. RF has 40 estimators, 3 maximal fea-tures, the 'gini' criterion, no maximum depth, and a random state of 42. Logistic Re-gression has a maximum iteration of 8000, the 'l2' penalty, fit_intercept set to True, the 'lbfgs' solver, and a random state of 42. Finally, the Bayes employs default parameters, with priors set to None and var_smoothing set to 1e-09. .” Refer to Section 5 Experimental Results & DiscussionsAnalysis (Lines 522-535) |
12.The evaluation metrics are mentioned, but there is no discussion on why these particular metrics were chosen and how they effectively measure the performance of the models. |
Adjustments were made in accordance with the recommendations, with an explanation provided in Section 4.5 Evaluation Model (Lines 464- 474). |
13. The presentation of results is somewhat disorganized and lacks a clear structure. |
To improve the clarity and structure of this research article, we followed the template journal format and included three new sections: Related Work, Experimental Setup, and Experimental Results. This change is intended to provide a more ordered and understandable presentation of the research findings. |
14.There is an overemphasis on the performance of certain models without a balanced discussion of their limitations or potential biases. |
We meticulously tested each algorithm at every stage, resulting in a wealth of experimental results. This comprehensive approach is intended to serve as valuable information and guidance for fellow researchers in their future applications. Our objective is to assess the efficiency of each algorithm and subsequently fine-tune machine learning algorithms for optimal performance. |
15.The discussion does not sufficiently compare the study's findings with existing literature or similar studies. There is a lack of critical analysis regarding the practical applicability of the findings, especially in real-world intrusion detection scenarios. |
We conducted experiments using established algorithms and compared the results to existing literature, as detailed in Section 5, Experimental Results & Discussions. Table 6 (Line 651-653) summarizes the comparison analysis. |
16. The conclusion is overly optimistic and does not adequately reflect the limitations and scope of the study. Future research directions are mentioned, but they are vague and lack specificity.
|
Make corrections. Additional suggestions section 6. Conclusions (Line 667-680)
|
Reviewer 2 Report
Comments and Suggestions for Authors
Minor issues:
In the Abstract, line 17, the text “feature importance is reduced” seems that is not precise. I suppose that is the number of features is what is reduced, not the importance.
In the Abstract, line 21, the word “Surprisingly” could be removed. In further chapters could extend this idea and justify way this result is unexpected.
In Keywords some of them are start with uppercase and others in lowercase. Please uniform the style.
The beginning of paragraph 4.1.1 is confusing. How many features have originally the dataset? What means “normalize” the dataset? Please rewrite it. In line 216 it is said that the number of characteristics is 69. Are these numbers coherent?
PCA is not a Feature Selection technique, but a Feature Reduction technique. Change it all over the text and in figure 1.
In line 288 the word “clusters” is not typically used. You can use class or category. The sentence contains redundant information or is somewhat incomprehensible.
In line 369 is not “Nave” but “Naive”
Mayor issues:
When listing the machine learning systems, it doesn't specify the hyperparameters that have been used or how they have been tuned. Please indicate which hyperparameters have been used for the various systems.
In line 429 it is said that the average accuracy is calculated using “cross-validation”, but the use of cross-validation is transversal to de measure, that is, can be applied to all measures. So, what strategy of generating train/test is used for the measures?
Author Response
Dear Reviewer
I aim to present the results of the improvements made, incorporating the suggestions provided in the aforementioned revisions. Throughout this process, careful consideration has been given to the proposals made by another reviewer, resulting in a collaborative effort to refine and enhance the content. The consolidated outcomes are reflected in the table presented below:
Comments and Suggestions for Authors |
Operation |
In the Abstract, line 17, the text “feature importance is reduced” seems that is not precise. I suppose that is the number of features is what is reduced, not the importance. |
Modifications have been made according to the first reviewer's suggestion, by reviewing and adjusting the description in the abstract according to the second reviewer's suggestion at the same time. Shown is the edited research article (Lines 13-22). |
In the Abstract, line 21, the word “Surprisingly” could be removed. In further chapters could extend this idea and justify way this result is unexpected. |
Modifications have been made according to the first reviewer's suggestion, by deleting the word “Surprisingly”. The description in the abstract section has been reviewed and adjusted according to the second reviewer's suggestion at the same time (Lines 13-22). |
In Keywords some of them are start with uppercase and others in lowercase. Please uniform the style. |
Modifications have been made according to suggestion, Improvements have been made by continuing to standardize the style by capitalizing keywords and using lowercase for others. As “Keywords: Intrusion Detection System; Machine Learning Techniques; Exploratory Data Analysis; Performance Evaluation; Feature Selection; CSE-CIC-IDS-2018 Dataset; Three Phase Models;” (Lines 23-24) |
The beginning of paragraph 4.1.1 is confusing. How many features have originally the dataset? What means “normalize” the dataset? Please rewrite it. In line 216 it is said that the number of characteristics is 69. Are these numbers coherent? |
Make changes in response to the reviewer's suggestions, including a more specific explanation of the dataset used in the experiment. As “After deleting the unnecessary features, the dataset is more usable for classification applications. The experiment will yield 8,997,323 rows of data and 69 characteristics.” Refer to Section 4.2.2 Exploratory Data Analysis (Lines 211-213) |
PCA is not a Feature Selection technique, but a Feature Reduction technique. Change it all over the text and in figure 1. |
Modifications have been made according to suggestion in Figure 1 (Lines 201-202), change the phrase "Feature Selection Technique" to "Feature Reduction Technique." |
In line 288 the word “clusters” is not typically used. You can use class or category. The sentence contains redundant information or is somewhat incomprehensible. |
Enhancements have been made in response to the reviewer's suggestions. The term "clusters" has been substituted with "class," and superfluous text has been eliminated to avoid repetition. As “Classification predicts data classes, and in the context of an Intrusion Detection System (IDS), attacks are categorized as binary or multiclass to discern benign or malicious network traffic. Binary classification involves two classes, while multiclass datasets can have n classes.” (Lines 299-301) |
In line 369 is not “Nave” but “Naive” |
Enhancements have been made in response to the reviewer's suggestions. Conduct a review and correct the accurate term "Naive" on line 369 and throughout the text per the reviewer's instructions. Update in Lines 378, 382, 383 |
When listing the machine learning systems, it doesn't specify the hyperparameters that have been used or how they have been tuned. Please indicate which hyperparameters have been used for the various systems. |
Enhancements have been made in response to the reviewer's suggestions. Continue by explaining the necessary parameter values for each categorization algorithm, As “Popular classification algorithms such as XGBoost, CART, DT, KNN, MLP, RF, LR, and Bayes are setup with precise parameter settings as follows. Key parameters for XGBoost include a learning rate of 0.2, 1000 estimators, a maximum depth of 5, and other parameters such as min_child_weight, subsample, and colsample_bytree set to 1. CART uses requirements like squared error, no maximum depth, a minimum sample split of three, and a minimum sample leaf of one. DT and CART have comparable characteristics, although KNN has 3 neighbors, uniform weights, the 'auto' algorithm, and a leaf size of 30. MLP employs (100, 50) hidden layer sizes, 1000 maximum itera-tions, 'relu' activation, and a random state of 42. RF has 40 estimators, 3 maximal fea-tures, the 'gini' criterion, no maximum depth, and a random state of 42. Logistic Re-gression has a maximum iteration of 8000, the 'l2' penalty, fit_intercept set to True, the 'lbfgs' solver, and a random state of 42. Finally, the Bayes employs default parameters, with priors set to None and var_smoothing set to 1e-09. .” Refer to Section 5 Experimental Results & DiscussionsAnalysis (Lines 522-535) |
In line 429 it is said that the average accuracy is calculated using “cross-validation”, but the use of cross-validation is transversal to de measure, that is, can be applied to all measures. So, what strategy of generating train/test is used for the measures? |
Modifications have been made according to suggestion by Rewrite and remover average accuracy according to the second reviewer's suggestion at the same time. (Lines 438-445) |
Reviewer 3 Report
Comments and Suggestions for Authors
The paper focuses on optimizing intrusion detection systems by exploring the impact of feature selection, normalization, and precision on the CSE-CIC-IDS-2018 dataset. The authors used various techniques to prepare the data for analysis across different classifiers, including feature scaling, feature selection, and normalization. The paper also discusses how feature importance was reduced to improve processing speed and decrease model complexity. The authors used three-phase precision to evaluate the performance of the intrusion detection system, and the results showed that the proposed system achieved high accuracy and low false positive rates. In terms of security, the paper emphasizes the importance of intrusion detection systems in protecting networks from cyber attacks. The authors also discuss the limitations of the proposed system and suggest future research directions. Overall, this paper provides valuable insights into optimizing intrusion detection systems using machine learning techniques. It highlights the importance of feature selection, normalization, and precision in improving the performance of intrusion detection systems. The proposed system has the potential to enhance network security and protect against cyber attacks. However, major revisions are needed before it is finally accepted.
In the comparative analysis presented in Table 5 of this paper, it is evident that the method proposed by S. Ullah et al. [14] outperforms the two methods introduced in this paper regarding CPU time (s) and accuracy. However, the authors fail to provide a detailed explanation regarding the specific advantages of the methods proposed in this paper compared to those of S. Ullah et al. [14]. It is essential to elucidate the superior attributes of the methods presented in this paper relative to S. Ullah et al. [14] to convincingly guide readers on the rationale for choosing the proposed approach. Therefore, it is recommended to enhance the comparative analysis to articulate each research method's distinctions and advantages clearly.
Author Response
Dear Reviewer
I aim to present the results of the improvements made, incorporating the suggestions provided in the aforementioned revisions. Throughout this process, careful consideration has been given to the proposals made by another reviewer, resulting in a collaborative effort to refine and enhance the content. The consolidated outcomes are reflected in the table presented below:
Comments and Suggestions for Authors |
Operation |
In the comparative analysis presented in Table 5 of this paper, it is evident that the method proposed by S. Ullah et al. [14] outperforms the two methods introduced in this paper regarding CPU time (s) and accuracy. However, the authors fail to provide a detailed explanation regarding the specific advantages of the methods proposed in this paper compared to those of S. Ullah et al. [14]. It is essential to elucidate the superior attributes of the methods presented in this paper relative to S. Ullah et al. [14] to convincingly guide readers on the rationale for choosing the proposed approach. Therefore, it is recommended to enhance the comparative analysis to articulate each research method's distinctions and advantages clearly. |
We appreciate the valuable feedback provided by the reviewer and have diligently incorporated the suggested adjustments into our manuscript. Additionally, we have taken into account comments from other reviewers to further enhance the quality of our research.
In response to your specific suggestions, we have elaborated on the nuances of our findings, particularly in relation to CPU runtime and accuracy, as outlined in Lines 641-648, As “Our findings differ from those of S. Ullah. et al. [14], particularly in terms of CPU runtime and accuracy. They applied the SMOTE balance strategy, which resulted in a smaller dataset that was only focused on CPU runtime during training in a binary class environment. In contrast, we used the entire dataset for training and testing without using separate balance methods, multiple classification classes, or reporting combined CPU runtimes. These methodological variations are most likely responsible for the disparities in CPU runtime and accuracy results between our study and theirs.” |
Best regards,
Mr. Surasit Songma
Round 2
Reviewer 1 Report
Comments and Suggestions for Authors
The authors have revised the paper well.
Author Response
Dear Reviewer
Thank you very much
Best regards,
Mr.Surasit Songma
Reviewer 2 Report
Comments and Suggestions for Authors
The authors have successfully addressed the issues I raised as a reviewer.
Author Response

(The authors gave the same response as above.)

Reviewer 3 Report
Comments and Suggestions for Authors
The authors have answered all the comments in the reviews. I am satisfied with their answers and, hence, now I consider the work can be accepted as is.